# Mineral Content and Volatile Profiling of *Prunus avium* L. (Sweet Cherry) By-Products from Fundão Region (Portugal)

**DOI:** 10.3390/foods11050751

**Published:** 2022-03-04

**Authors:** Ana R. Nunes, Ana C. Gonçalves, Edgar Pinto, Filipa Amaro, José D. Flores-Félix, Agostinho Almeida, Paula Guedes de Pinho, Amílcar Falcão, Gilberto Alves, Luís R. Silva

**Affiliations:** 1CICS-UBI—Health Sciences Research Centre, University of Beira Interior, Av. Infante D. Henrique, 6200-506 Covilha, Portugal; araqueln@gmail.com (A.R.N.); anacarolinagoncalves@sapo.pt (A.C.G.); jdflores@usal.es (J.D.F.-F.); 2CNC—Centre for Neuroscience and Cell Biology, Faculty of Medicine, University of Coimbra, 3004-504 Coimbra, Portugal; 3Laboratory of Pharmacology, Faculty of Pharmacy, University of Coimbra, Azinhaga de Santa Comba, 3000-548 Coimbra, Portugal; acfalcao@ff.uc.pt; 4LAQV/REQUIMTE, Department of Chemical Sciences, Faculty of Pharmacy, University of Porto, 4050-313 Porto, Portugal; ecp@ess.ipp.pt (E.P.); aalmeida@ff.up.pt (A.A.); 5Department of Environmental Health, School of Health, Polytechnic Porto, 4200-072 Porto, Portugal; 6UCIBIO, REQUIMTE, Laboratory of Toxicology, Faculty of Pharmacy, University of Porto, 4050-313 Porto, Portugal; filipa_amaro@hotmail.com (F.A.); pguedes@ff.up.pt (P.G.d.P.); 7Associate Laboratory i4HB, Institute for Health and Bioeconomy, Faculty of Pharmacy, University of Porto, 4050-313 Porto, Portugal; 8CIBIT—Coimbra Institute for Biomedical Imaging and Translational Research, University of Coimbra, Azinhaga de Santa Comba, 3000-548 Coimbra, Portugal; 9CPIRN-UDI-IPG—Center of Potential and Innovation of Natural Resources, Research Unit for Inland Development, Polytechnic Institute of Guarda, 6300-559 Guarda, Portugal

**Keywords:** *Prunus avium* L., bio-residues, volatile compounds, essential elements, HS-SPME/GC-MS, ICP-MS

## Abstract

Large amounts of *Prunus avium* L. by-products result from sweet cherry production and processing. This work aimed to evaluate the mineral content and volatile profiling of the cherry stems, leaves, and flowers of the Saco cultivar collected from the Fundão region (Portugal). A total of 18 minerals were determined by ICP-MS, namely 8 essential and 10 non-essential elements. Phosphorus (P) was the most abundant mineral, while lithium (Li) was detected in trace amounts. Three different preparations were used in this work to determine volatiles: hydroethanolic extracts, crude extracts, and aqueous infusions. A total of 117 volatile compounds were identified using HS-SPME/GC-MS, distributed among different chemical classes: 31 aldehydes, 14 alcohols, 16 ketones, 30 esters, 4 acids, 4 monoterpenes, 3 norisoprenoids, 4 hydrocarbons, 7 heterocyclics, 1 lactone, 1 phenol, and 2 phenylpropenes. Benzaldehyde, 4-methyl-benzaldehyde, hexanal, lilac aldehyde, and 6-methyl-5-hepten-2-one were the major volatile compounds. Differences in the types of volatiles and their respective amounts in the different extracts were found. This is the first study that describes the mineral and volatile composition of Portuguese sweet cherry by-products, demonstrating that they could have great potential as nutraceutical ingredients and natural flavoring agents to be used in the pharmaceutical, cosmetic, and food industries.

## 1. Introduction

In recent years, bioactive compounds of natural origin have been arousing great interest due to their potential for the prevention and treatment of several diseases, as well as for health promotion [1]. The richness of plants in phenolic compounds makes them an excellent and low-cost source of phytochemicals with wide applicability in pharmaceutical, food, and cosmetic fields [2,3].

In addition to phenolics, other bioactive compounds can be obtained from plants, fruits, and their by-products, such as minerals, vitamins, and volatile substances [4,5,6]. Minerals are one of the most important constituents of the diet. They have a key role in the structural components of the human body, such as bones and teeth, and are involved in many metabolic processes [7]. Although the red fruits (e.g., cherries and raspberries) are recognized as a source of essential elements, such as potassium (K), calcium (Ca), phosphorus (P), and magnesium (Mg) [5,8], there are other parts of the fruit that can also contain these elements. According to García-Aguilar et al. [9], black cherry seeds are a promising source of mineral elements with bioactive properties.

Volatile organic compounds (VOCs) represent an important part of the plant metabolome, which act, either independently or in combination, to produce a characteristic aroma. In plants, VOCs are also involved in biotic communications and in abiotic stress acclimation [4]. The VOC profiling associated with sensory tools can be useful, for example, in the identification of essential compounds to describe the quality of products [10] or to assess the authenticity of flavoring substances [11]. In fact, food acceptance is directly related to its flavor [11]. Furthermore, these compounds have been shown to be able to improve the shelf-life and safety of fresh and processed fruits [12] as well as industrially produced foods, maintaining their aroma after processing and storage [13]. Regarding the biological potential of VOCs, several studies have reported the antimicrobial [14], antiproliferative [15], and antioxidant properties [16] of these compounds. It is important to note that the variability of these compounds in plants is due to several factors, such as the climate conditions, geographical location, harvesting process, maturation state, time, and storage conditions, among others [5].

According to the literature, more than 20% of fruits and vegetables are transformed into waste [17], and the scientific community has developed an interest in these bio-residues, as they represent a source of functional substances whose recovery can generate economic value, contributing to waste reduction [2,18] and the circular economy. *Prunus avium* L., sweet cherry, is a member of the Rosaceae family and is a well-known and very popular fruit worldwide due to its organoleptic properties and nutritional or health benefits [3,19]. According to the most recent data of the U.S. Department of Agriculture, cherry production in the United States during the year 2020 was around 265 million tons [20]. In Portugal, particularly in the Cova da Beira region (Fundão), cherry cultivation is of great importance for the local economy where the agronomic conditions are very favorable for its production [21]. In 2020, between 3500 and 7000 tons of cherries were produced in this region. Fresh cherry consumption and industrialization (e.g., jams and liquors) generate a variety of bio-residues, such as seeds, skin, and pulp, which are then discarded [22]. In addition, stems, leaves, and flowers are also produced, with spring and autumn being responsible for the most abundant accumulation of flowers and leaves. Thus, large amounts of sweet cherry by-products are generated, which can be valued from a biological and functional point of view.

To the best of our knowledge, data on the mineral content and volatile composition of *P.*
*avium* stems, leaves, and flowers of the Saco cultivar from the Fundão region (Portugal) have not been described. Saco is one of the oldest and most important cultivars in this region [19]. In a previous paper [22], we reported the phenolic profile and some of its bioactive properties, namely the antioxidant and antiproliferative activities, of the by-products studied in the present work. In this context, this study aimed to provide more detailed information on the mineral and volatile composition of sweet cherry stems, leaves, and flowers. Knowledge of the phytochemical composition of these by-products is essential to assess their potential interest as a source of bioactive compounds and their applicability in the industry. Among the various possibilities, some minerals and VOCs found in *P.*
*avium* by-products might be incorporated in nutraceuticals, natural products, drugs, and cosmetics in the food and pharmaceutical industries, contributing to the circular economy.

## 2. Materials and Methods

### 2.1. Standards and Reagents

All chemicals were of analytical grade and were purchased from Sigma Aldrich (St. Louis, MO, USA) unless otherwise stated. All solutions were prepared using ultrapure water (>18.2 MΩ.cm at 25 °C) obtained with a Milli-Q RG (Millipore, Billerica, MA, USA) water purification system. For the ICP-MS analysis, the AccuTrace™ (AccuStandard^®^, New Haven, CT, USA) ICP-MS-200.8-IS-1 (100 μg/mL of Sc, Y, In, Tb, and Bi) solution was used to prepare the internal standard solution. The calibration standards were prepared from the 10 μg/mL multi-element ICP-MS standard solution (PlasmaCAL SCP-33-MS, SCP Science, Clark GrahamAve, Baied’Urfé, Quebec, Canada), from the 100 μg/mL multi-ion chromatography standard 3 solution (AccuSPEC, SCP Science, Baied’Urfé, QC, Canada), and from the 1000 μg/mL Hg standard solution (TraceCERT^®^, Sigma-Aldrich, Buchs, Switzerland). For the GC-MS analysis, the standard compounds were purchased from various suppliers: hexanal, benzaldehyde, and phenylethyl alcohol were obtained from SAFC (Steinheim, Germany); 1-hexanol was obtained from Fluka (Buchs, Switzerland); heptanal, octanal, 3-methyl-1-butanol, 2-methyl-1-butanol, linalool, methyl salicylate, methyl hexanoate, *β*-cyclocytral, nonanal, 3-methyl-butanal, 2-methyl-butanal, hexanal, (*E*)-2-hexenal, (*E*)-2-hexen-1-ol, heptanal, 2-methyl-1-butanol, benzyl alcohol, ethyl acetate, acetic acid, octanoic acid, decanoic acid, and α-terpineol were purchased from Sigma–Aldrich (St. Louis, MO, USA).

### 2.2. Plant Material and Extract Preparation

Stems, leaves, and flowers of *P. avium* were collected during the months of April and June 2018 by a local producer in the Fundão region, Cimo da Aldeia–Pêro Viseu (40°12′36″ N, 7°26′21″ W, Cova da Beira, Portugal). Subsequently, the samples were transported, frozen, and stored in our laboratory of the Health Sciences Research Centre of the University of Beira Interior, Portugal (CICS-UBI), as previously described [22].

The hydroethanolic extracts and aqueous infusions were prepared as described [22]. In brief, the hydroethanolic extracts were prepared using 1 g of sample with ethanol and water (50:50, *v*/*v*) and were processed via maceration, sonication, and filtration. For the aqueous infusions, 1 g of the samples were subjected to infusion in 100 mL of water. The crude extract was freeze dried (Figure 1). The extraction yields from dry material have been previously reported [22].

### 2.3. Mineral Analysis

#### 2.3.1. Sample Digestion

Acid digestion of the samples was performed in a Milestone (Sorisole, Italy) MLS 1200 Mega high-performance microwave digestion unit equipped with an HPR-1000/10 S rotor. About 250 mg of lyophilized powdered sweet cherry stems, leaves, and flowers were weighted directly into the microwave PTFE vessels. Then, 3 mL of 69% (*w*/*w*) HNO_3_ and 1 mL of 30% (*v*/*v*) H_2_O_2_ were added to each vessel, and the mixture was subjected to a microwave heating program as described by Gonçalves et al. [5]. After cooling, the vessel contents were transferred to 25 mL volumetric flasks that were topped up with ultrapure water. BCR-679-certified reference material (white cabbage, supplied by the EC Institute for Reference Materials and Measurements, Geel, Belgium) was used for analytical quality control.

#### 2.3.2. ICP-MS Analysis

The analysis of the solutions was performed according to previous works [5,23]. Calcium (Ca), potassium (K), sodium (Na), and manganese (Mn) were determined by flame atomic absorption spectrometry (FAAS) using a PerkinElmer (Waltham, MA, USA) AAnalyst 200 instrument. External calibration was carried out as described by Gonçalves et al. [5]. The phosphomolybdate method was used to determinate the total phosphorus (P) according to Murphy and Riley [24]. Briefly, ammonium molybdate and antimony potassium tartrate reacted in acid medium with the orthophosphate from the samples to form the heteropoly acid phosphomolybdic acid. This was then reduced by ascorbic acid to produce a blue color that was measured spectrophotometrically at 880 nm [5].

Inductively coupled plasma-mass spectrometry (ICP-MS) was used to determine the other elements using an iCAP™ Q instrument (Thermo Fisher Scientific, Bremen, Germany) equipped with a MicroMist nebulizer, a baffled cyclonic spray chamber (Peltier-cooled), a standard quartz torch, and a two-cone interface (sample and skimmer cones). High-purity argon (Gasin II, Leça da Palmeira, Portugal) was used as the nebulizer and plasma gas. The ICP-MS instrument operational parameters used are described in the study developed by Gonçalves et al. [5]. ^7^Li, ^9^Be, ^27^Al, ^51^V, ^52^Cr, ^55^Mn, ^57^Fe, ^59^Co, ^60^Ni, ^65^Cu, ^66^Zn, ^75^As, ^82^Se, ^85^Rb, ^88^Sr, ^111^Cd, ^118^Sn, ^121^Sb, ^137^Ba, ^205^TI, ^208^Pb, and ^209^Bi were the elemental isotopes (*m*/*z* ratio) monitored for the analytical determinations. The elemental isotopes ^45^Sc, ^89^Y, ^115^In, ^159^Tb, and ^209^Bi were used as internal standards. The instrument was tuned for maximum signal sensitivity, stability, low oxides, and doubly charged ion formation using the Tune B iCAP Q solution (Thermo Fisher Scientific; 1 μg/L of Ba, Bi, Ce, Co, In, Li, and U in 2% HNO_3_ plus 0.5% HCl). The limits of detection (LOD) and quantification (LOQ) were calculated as the concentration corresponding to 3.3 and 10 times the standard deviation of 10 replicate measurements of the calibration blank (2% HNO_3_), respectively (Appendix A). All samples were analyzed in triplicate. The results were expressed as mg/kg on a dry weight (dw) basis.

### 2.4. Volatile Organic Compound (VOC) Analysis

#### 2.4.1. Volatile Extraction by Headspace Solid-Phase Microextraction (HS-SPME)

For the analysis of VOCs (aldehydes, alcohols, acids, ketones, esters, acids, monoterpenes, norisoprenoids, heterocyclics, hydrocarbons, phenols, and phenylpropenes), the HS-SPME with a 50/30 μm divinylbenzene/carboxen/polydimethylsiloxane (DVB/CAR/PDMS) (Supelco Bellefonte, PA, USA) was used as described in previous studies, with minor modifications [5,23]. Approximately 0.1 g of lyophilized hydroethanolic extract, aqueous infusion, and crude extract of *P. avium* stems, leaves, and flowers were transferred to a 10 mL glass vial with 2 mL of H_2_O and 0.43 g of NaCl (salting out effect) and stirred (500 rpm) for 5 min. Then, the fiber was exposed to the headspace for 10 min under continuous agitation (250 rpm) at 45 °C. The analytes were then thermally desorbed in the injector of an EVOQ 436 GC system (Bruker Daltonics, Fremont, CA, USA) coupled to a SCION SQ mass detector and a Bruker Daltonics MS workstation (software version 8.2) coupled to a Combi-PAL autosampler.

#### 2.4.2. Gas Chromatography-Mass Spectrometry (GS-MS) Analysis

The volatile profile was obtained according to the method described by Gonçalves et al. [5] using GC-MS. The analysis was performed using a capillary column Rxi-5Sil MS (30 m × 0.25 mm × 0.25 µm) from RESTEK (Bellefonte, PA, USA). High-purity He C-60 (Gasin, Portugal) was used as the carrier gas at a constant flow rate of 1.0 mL min^−1^. The oven temperature increased from 40 °C to 250 °C with a speed of 5 °C/min, followed by an increase from 5 °C/min to 300 °C. The injection was conducted in splitless mode, and the injector was at 250 °C (held for 20 min). The MS detector was operated in EI mode, and data acquisition was performed in full scan mode with a mass range between 40 and 500 *m*/*z* at a scan rate of 6 scans/s. The analysis was performed in full scan mode. The VOCs were identified according to their retention times. Kovats indices were also used to compare the detected volatiles and standard compounds with data from the literature [25]. A comparison of the MS fragmentation patterns with those of pure compounds analyzed under the same conditions and mass spectrum database searches was performed using the National Institute of Standards and Technology (NIST) MS 14 spectral database [25].

### 2.5. Statistical Analysis

All assays were performed in triplicate in three independent experiments. All experimental results were reported as means ± standard deviations. One-way ANOVAs with Tukey’s multiple comparison tests were used for statistical comparisons between the different sweet cherry by-products using GraphPad Prism 8.4 software. *p*-value < 0.05 was accepted as denoting statistical significance. A principal component analysis (PCA) was performed using XLSTAT 2020 software (Addinsoft, New York, NY, USA). This method has been shown to be effective in detecting similarities between data sets, making it possible to identify the variables that determine the similarities.

## 3. Results and Discussion

### 3.1. Mineral Content

Minerals (inorganic elements) can be divided into macrominerals and microminerals (trace elements) and also into essential and non-essential minerals [26]. These elements play vital roles in the body, favoring the balance and maintenance of different metabolic processes. *P. avium* fruits have a great diversity of essential minerals, making them suitable for a healthy diet [5,27]. According to the study by Gonçalves et al. [5], Portuguese sweet cherries have high levels of potassium (K), phosphorus (P), and magnesium (Mg).

The mineral content in the *P. avium* stems, leaves, and flowers are shown in Table 1 and Table 2. In total, 18 minerals were detected in these sweet cherry by-products, 8 of which are essential elements (Table 1), and 10 are non-essential elements (Table 2). The total mineral content was 1587 mg/kg dw in stems, 1341 mg/kg dw in leaves, and 960 mg/kg dw in flowers. As expected, the cherry by-products of the Saco cultivar had lower total mineral content compared to the cherry fruit of the same cultivar [5].

#### 3.1.1. Essential Elements

Table 1 shows the content of the macrominerals phosphorus (P) and sodium (Na) and the trace elements manganese (Mn), zinc (Zn), iron (Fe), cobalt (Co), copper (Cu), and selenium (Se) in the stems, leaves, and flowers of sweet cherry. These elements are necessary for the proper functioning of the body. The element found in the highest content in all analyzed cherry by-products was P, followed by Na in the stems, Mn in the leaves, and Zn in the flowers (Table 1). In general, the results obtained are similar to the data found in *Prunus persica* by-products [4] and *P. avium* fruits [5].

Phosphorus was the major element in all cherry by-products, ranging from 837 to 1345 mg/kg dw, with *P. avium* stems presenting the highest levels of this mineral (Table 1). In fact, the content of P found in the stems was significantly higher than in the leaves and the flowers (Table 1). This element is present in fruits in the range of 9.9–94.3 mg/100 g [28]. When compared to other studies, *P. avium* by-products contain higher levels of P than sweet cherries, which have average values around 1168 mg/kg dw [5]. This essential mineral is involved in regulatory functions in the body. It is part of bones and teeth, and contributes to the metabolic components, as it is involved in ATP and GTP production and calcium (Ca) homeostasis [28].

In addition, low amounts of Na were detected in *P. avium* stems (Table 1). This result is in agreement with the literature, which reports that Na is present in raw vegetables at relatively low levels [28]. However, cherry stems have lower levels of Na (79.5 mg/kg dw) when compared to *P. avium* fruit (208 mg/kg dw) [5].

The essential trace elements Co, Cu, Fe, Mn, Se, and Zn were quantified in cherry by-products (Table 1). In general, the obtained content of Fe in the stems, leaves, and flowers (around 32.5, 66.8, and 30.9 mg/kg dw, respectively) was higher than in cherries of the same cultivar (around 2.44 mg/kg dw) [5]. Copper was found in the stems in greatest amount, followed by the flowers and leaves (Table 1). On the other hand, the leaves showed higher levels of Mn, while the flowers were richer in Zn (Table 1). While Fe is involved in oxygen transport and integrates proteins and metabolic enzymes (e.g., catalase), Mn possesses antioxidant activity in the mitochondria, and it is involved in bone development and wound healing. In plants, namely in leaves, Mn is an essential mineral for photosynthesis [29]. Zn is necessary in many metabolic processes and is involved in growth, the immune response, neurological function, and reproduction. Co was detected only in *P. avium* leaves (Table 1).

#### 3.1.2. Non-Essential Elements

In this work, 10 non-essential elements (aluminum (Al), arsenic (As), barium (Ba), cadmium (Cd), chromium (Cr), lithium (Li), nickel (Ni), lead (Pb), rubidium (Rb), and Sr (strontium)) were found in the stems, leaves, and flowers of *P. avium* (Table 2). According to Gonçalves et al. [5], other elements, such as beryllium (Be), vanadium (V), and silver (Ag), were quantified in by-products from cherries of the same variety, but in the present study, these trace elements were not detected.

In general, the non-essential mineral found in the highest amount in the cherry by-products was Ba, followed by Al > Sr > Rb > Pb > Li (Table 2). Significant differences were found between the studied by-products (Table 2). The leaves and stems were the samples that showed the highest amounts of non-essential elements, whereas the flowers presented the lowest levels.

Al and Ba were quantified in all samples but in major amounts in the leaves and stems, respectively (Table 2). Compared to the cherry fruit of the Saco cultivar, their by-products are richer in these non-essential elements [5] but with lower levels than strawberries and mulberries [30]. Regarding the other non-essential elements, Rb and Sr were also found in the stems, leaves, and flowers (Table 2). Sr was detected in significant amounts in the leaves and stems (20.3 ± 1.2 and 18.62 ± 0.53 mg/kg dw, respectively). According to the literature, this element can be efficiently accumulated by plants and presents high physical and chemical similarities with Ca, which is essential for plants [31]. These elements were not found in *P. avium* fruits [5]. In addition, low levels of Li and Pb (<1 mg/kg dw) were found in the cherry by-products (Table 2).

Due to the high demand for bioactive compounds from plants and their by-products by the food, pharmaceutical, and cosmetic industries, it is essential to characterize their safety profile not only for the consumer but also regarding environmental toxicity. The accumulation of toxic elements in the human body leads to an increase in free radical production and, consequently, to oxidative stress and the development of several diseases [5]. To the best of our knowledge, this study is the first report on the mineral content of sweet cherry by-products from the Fundão region, Portugal.

### 3.2. VOC Profiles

The volatile composition of the three different types of extracts (hydroethanolic, aqueous, and crude) of *P. avium* stems, leaves, and flowers is shown in Table 3. In this study, the HS-SPME/GC-MS allowed the identification and semi-quantification of 117 VOCs belonging to different chemical classes: 31 aldehydes, 14 alcohols, 16 ketones, 30 esters, 4 acids, 4 monoterpenes, 3 norisoprenoids, 4 hydrocarbons, 7 heterocyclics, 1 lactone, 1 phenol, and 2 phenylpropenes. Some of these compounds have previously been reported in Portuguese sweet cherries [5].

In general, we can observe that *P. avium* stems, leaves, and flowers have similar volatile composition for most of the identified classes but with some differences. The main classes found in the *P. avium* by-products were aldehydes, alcohols, esters, and ketones (Figure 2). In this context, aldehydes represent 65.8% of the VOCs identified in all extracts of the by-products, while alcohols, esters, and ketones represent 12.7%, 8.2%, and 8.1%, respectively (Figure 2). The other identified classes represent a relative percentage of VOCs that ranges between 0.13% and 1.4% (Figure 2).

Stem hydroethanolic extracts showed the highest diversity of aldehydes (17 of 31 compounds), while flower hydroethanolic extracts exhibited the highest number of esters (18 of 30 compounds). Ketones were found mainly in the stem crude extract and leaf and flower aqueous infusions (6 of 16 compounds). On the other hand, the stem and leaf aqueous infusions presented the greatest variety of alcohols (7 of 14 compounds).

Among all the extracts analyzed, the greatest diversity of compounds was found in the hydroethanolic extract and in aqueous infusion of flowers (42 and 40 compounds, respectively), followed by the leaf aqueous infusion (38 volatile compounds) (Table 3). The least complex volatile composition was found in the leaf hydroethanolic extract and the flower crude extract (27 and 26 compounds, respectively) (Table 3).

#### 3.2.1. Aldehydes

The aldehydes in cherry stems, leaves, and flowers account for about 65.8% of the total VOCs identified in this study (Figure 2). They were essentially found in the stem hydroethanolic extract (17 in 31), the leaf aqueous infusion (14 in 31), and the crude extract of flowers (14 in 31) (Table 3). In Figure 3A, we can see that aldehydes represent about 44.8% of the total VOCs found in the stem hydroethanolic extract, followed by the crude extract of flowers (39.9%) (Figure 3C).

Benzaldehyde (14), 4-methyl-benzaldehyde (18), (*E*)-2-hexenal (8), hexanal (6), (*Z*)-2-heptenal (13), and nonanal (19) were the main aldehydes found in the *P. avium* by-products (Table 3). The presence of these compounds was reported in *P. avium* fruit in previous studies [5,32,33,34]. Notably, benzaldehyde was found in all cherry by-product extracts in significant amounts (Table 3). These results are in agreement with the study by Zhang et al. [35], who detected this compound and (*E*)-2-hexenal in the sweet cherry flowers of cultivars Brooks, Black Pearl, Tieton, and Summit. This benzaldehyde has a sweet, floral, and spice-like odor and is widely used by the cosmetic and food industries as an aliphatic fragrance and flavor material [36]. The presence of benzaldehyde in *P. avium* fruits and in *P. genus* has already been reported in previous works [5,33,37]. Nonanal and hexanal have also been shown to have significant antimicrobial activity [38,39].

Safranal (26) is a plant secondary metabolite isolated from saffron and was found in an aqueous infusion of *P. avium* leaves (Table 3). Promising pharmacological properties have been described for this compound, namely neuroprotective and anti-inflammatory activities [40,41]. No data could be found in the literature on the presence of safranal in *P. avium* fruits and their by-products.

Lilac aldehydes (20, 21, and 25) were found in all cherry flower extracts (Table 3). These compounds are olfactory molecules of great interest to pollinators, and they are commonly found in *P. avium* flowers [35], *Syringa vulgaris* L. [42], and bee pollen [43]. The lilac aldehydes in cherry flowers are characteristic aroma components [35].

#### 3.2.2. Alcohols

Alcohols are the second class of VOCs detected in greater amounts in *P. avium* by-products, representing about 12.7% of the total VOC content (Table 3 and Figure 2). These results agree with the previous data reported for cherries [5,32,34]. Flower and leaf aqueous infusions contain about 12.3% and 12.1%, respectively (Figure 3B,C).

The main alcohols found in the cherry by-products were 2-ethyl-1-hexanol (41), lilac alcohol (isomer C) (45), 1-hexanol (39), 1-octen-3-ol (40), and phenylethyl alcohol (43) (Table 3). These VOCs were detected mainly in the aqueous infusions of stems and leaves and also in the flower hydroethanolic extract. 2-Ethyl-1-hexanol was the alcohol present in large amounts in most *P. avium* by-product extracts (Table 3). Compared to other studies, 2-ethyl-1-hexanol was also the predominant alcohol found in cherries [5] and their flowers [35]. Phenylethyl alcohol was found mainly in stems and flowers, and it is an important flavoring ingredient. Moreover, this compound is one of the main aroma constituents of rose essential oil, and its fresh and sweet smell has calming effects as well as anti-inflammatory and antibacterial effects [35]. No alcohols were found in any of the sweet cherry leaf extracts (Table 3).

#### 3.2.3. Ketones

Ketones are typical aromatic components and were found in all analyzed extracts of the *P. avium* by-products (Table 3), corresponding to 8.1% of the total VOCs identified in this study (Figure 2). The aqueous infusion and the hydroethanolic extract of cherry leaves showed the highest diversity of ketones (17.9% and 9.2%, respectively, of the total VOC content in each extract) (Figure 3B), followed by the stem aqueous infusion (Figure 3A). 6-Methyl-5-hepten-2-one (53), acetophenone (57), and 1-octen-3-one (52) were the main ketones present in the cherry by-products. Their presence agrees with the results obtained in cherries [5,32,44]. 6-Methyl-5-hepten-2-one is an oxidative by-product or degradation product derived from licopene, farnesene, citral, or conjugated trienols [45] and has been detected mainly in leaf extracts (Table 3). This compound has already been described in different cultivars of sweet cherry fruit from the Fundão region (Portugal) [5] as well as in Chinese and Spanish cherries [33,46] and also in the stems of other plants [25].

#### 3.2.4. Esters

Esters are the class of VOCs that presented the greatest diversity of compounds, after aldehydes, in the *P. avium* stems, leaves, and flowers (Table 3). These compounds represented 8.2% of the total VOCs (Figure 2) and were found mostly in the leaf hydroethanolic extract, followed by the hydroethanolic extracts of flowers and stems (Table 3 and Figure 3). Most esters can impart plant fruit fragrances. Ethyl hexanoate (75), ethyl benzoate (79), and ethyl acetate (62) were the most relevant esters identified in this work. Among these esters, ethyl hexanoate seems to be the main ester responsible for the aroma of *P. avium* fruits [5]. This VOC has also been found in cheese, wine, and apples [47]. Esters have fruity flavor odors with sensory descriptions that range from fruity and pleasant (e.g., ethyl acetate), banana- and pear-like (e.g., isoamyl acetate), rose- and honey-like (e.g., ethyl phenylacetate), or apple-like and floral (e.g., ethyl hexanoate and ethyl octanoate). Isopropyl myristate (90) was found in all extracts of the *P. avium* by-products (Table 3). The literature data showed that this VOC has a high toxicity against gram-negative microorganisms, and it is being suggested as a solvent for ointments [48].

#### 3.2.5. Acids, Monoterpenes, Norisoprenoids, and Hydrocarbons

Organic acids are found at trace levels in plants but play an important role in plant primary metabolism, contributing to the regulation of plant development and growth. Among the acids identified in the *P. avium* by-products, acetic acid (92) and nonanoic acid (94) were the main ones (Table 3). Acetic acid was previously detected in cherries [5,32].

Monoterpenes, norisoprenoids, and hydrocarbons were also found in trace amounts, with a total content ranging between 0.73 and 1.25%. Among the monoterpenes, linalool (98) was found in significant amounts in the aqueous leaf and flower infusions (Table 3). In industry, this compound is applied in cosmetic flavors and food fruit flavors with antibacterial and antiviral effects [35]. Limonene (96) and linalool have already been described in sweet cherry flowers of different cultivars [35].

Regarding the norisoprenoids, they are the compounds responsible for the aromatic quality of *P. avium* fruits [5]. The norisoprenoids found in sweet cherry leaf extracts were *β*-cyclocitral (100), and *α*- and *β*-ionone (101 and 102) (Table 3). α-Ionone was also detected in the stem aqueous infusion. *β*-Cyclocitral is found primarily in plant roots and has been identified in tomatoes, rice, tea, grapes, various trees, and moss [49]. According to the literature data, this compound may be a beneficial strategy to enhance root growth and plant vigor in agriculture [49]. On the other hand, α-ionone may be involved in plant protection against herbivore attacks [50].

Styrene was the main hydrocarbon detected in the stems, leaves, and flowers of sweet cherry (Table 3) and has been identified as a natural constituent in a wide variety of foods and beverages [51]. Structurally, this compound is similar to many naturally occurring flavor molecules, such as cinnamic acid, cinnamic aldehyde, and cinnamyl benzoate, among others. Thus, it is possible that styrene can be produced during the biodegradation of those substances [51]. 2,4-Dimethyl-1-heptene (103), 1,3-dimethylheptane (104), and *o*-cymene (106) were also found in *P. avium* by-products (Table 3).

#### 3.2.6. Heterocyclics

Furanic compounds such as 2-methylfuran (107), methyl tetrahydrofuran (108), 2-ethylfuran (109), and 2-penthylfuran (113), were found in the *P. avium* leaf extracts (Table 3). It is possible that the thermal degradation of amino acids, thermal oxidation of ascorbic acid, and oxidation of polyunsaturated fatty acids may contribute to the formation of these VOCs [52]. These compounds have already been reported in wine by-products [47]. 2,3-Dimethylpyrazine (112) was identified in the flower aqueous infusion (Table 3). Pyrazine compounds are heterocyclic nitrogen with distinct organoleptic properties and have been identified in *Catharanthus roseus* leaves [25]. In addition, a study by Premkumar et al. [53] attributed some antimicrobial activities to pyrazines. No data could be found in the literature on these compounds in *P. avium* fruits.

#### 3.2.7. Lactones

4-Methyl-4-vinylbutyrolactone (114), also known as 4-hydroxy-4-methyl-5-hexenoic acid gamma lactone or 4-methyl-4-vinyl-1,4-butanolide, was the main lactone found in the hydroethanolic extract and aqueous infusion of cherry flowers (Table 3). However, little is known about biological activity of this compound.

#### 3.2.8. Phenols

4-Ethylphenol (115) was identified in the flower infusion (Table 3). Commonly, this VOC has a significant impact on the organoleptic properties of fermented foods, such as wine [54]. In the literature, there is no evidence of the presence of this compound in plants, namely in *P. avium* fruit and their by-products.

#### 3.2.9. Phenylpropenes

Phenylpropenes are a class of volatile compounds that function as floral attractants for pollinators and have antimicrobial and antifungal properties in plant defense against pathogens [55]. Although they contribute a small percentage to the total VOCs, these compounds are important in the flavor and aroma of herbs and spices. In this study, only two phenylpropenes were found in the crude extract of *P. avium* leaves: estragole (116) and apioline (117) (Table 3). The presence of estragole was detected in ripe apples [55]. On the other hand, apioline is usually found in the essential oil of *Apium graveolens* L. and in *Petroselinum crispum* [56]. As far as we know, it is the first report about the presence of these phenylpropenes in *P. avium* leaves.

Among the phenolic compounds described in *P. avium* stems, leaves, and flowers in the previous study [22] and the VOCs identified in the present work, it is important to know that there are other compounds, such as cyanogenic compounds, that can represent potential risks to consumers at doses between 0.5 and 3.5 mg hydrogen cyanide (HCN) per kg body weight [57]. They are common in Rosaceae family and release hydrogen cyanide when chewed or digested. However, cyanogenic compounds play crucial roles in the organization of the chemical defense system in plants and in plant–insect interactions [57]. Amygdalin is one of these compounds and recent studies have shown that it has anti-cancer effects [58].

### 3.3. Principal Component Analysis (PCA)

In order to obtain a general evaluation and comparison of the different extracts (hydroethanolic, aqueous infusion, and crude) of the *P. avium* stems, leaves, and flowers in terms of volatile composition, a PCA was performed. Figure 4 shows that 54.7% of the total variance of the data is represented by the principal axes F1 and F2. Of these two principal components, F1 described 31.1% of the total variation, and F2 explained 23.6%. In this context, the PCA explains the main differences between the different extracts of the sweet cherry by-products and groups them according to their quantitative levels of VOCs.

**Table 3 foods-11-00751-t003:** Volatile composition of extracts of *Prunus avium* L. by-products from the Saco variety from the Fundão region (Portugal).

										A (SD) ^c^						
							Stems			Leaves			Flowers			
Compounds		RT	Most Abundant Ions (*m*/*z*)	RI ^a^ (Reported)	RI ^b^ (Calculated)	Rmatch	Hydroethanolic Extract	Aqueous Infusion	Crude Extract	Hydroethanolic Extract	Aqueous Infusion	Crude Extract	Hydroethanolic Extract	Aqueous Infusion	Crude Extract	Sensorial Description
**Aldehydes**																
**1**	Crotonaldehyde *^L^*^2^	2.51	41/70	629	-	933	nd	nd	1.73 × 10^+08^ (1.54 × 10^+08^)	1.33 × 10^+09^ (5.35 × 10^+07^)	nd	nd	nd	nd	nd	
**2**	3-Methyl-butanal *^L^*^2^	2.56	44/58	652	-	653	nd	nd	3.04 × 10^+08^ (6.76 × 10^+07^)	nd	5.47 × 10^+08^ (1.01 × 10^+08^)	nd	1.60 × 10^+10^ (8.26 × 10^+07^)	3.16 × 10^+09^ (8.56 × 10^+08^)	7.59 × 10^+08^ (2.10 × 10^+07^)	Fatty, sour, and peach
**3**	2-Methyl-butanal *^L^*^2^	2.65	57	662	-	841	3.61 × 10^+08^ (5.54 × 10^+07^)	nd	4.88 × 10^+08^ (1.27 × 10^+08^)	7.07 × 10^+08^ (2.99 × 10^+08^)	1.27 × 10^+09^ (1.05 × 10^+08^)	5.93 × 10^+07^ (7.72 × 10^+06^)	2.62 × 10^+09^ (1.63 × 10^+08^)	3.43 × 10^+09^ (2.07 × 10^+09^)	1.21 × 10^+09^ (6.70 × 10^+07^)	Green, grass, and fruity
**4**	Pentanal *^L^*^2^	3.00	44/58	669	-	882	nd	nd	9.90 × 10^+08^ (9.38 × 10^+08^)	nd	nd	nd	6.76 × 10^+08^ (8.50 × 10^+08^)	nd	1.02 × 10^+09^ (5.00 × 10^+08^)	Almond, bitter, malt, oil, and pungent
**5**	3-Methyl-2-butenal *^L^*^2^	4.41	55/84	782	-	949	nd	nd	nd	nd	nd	3.90 × 10^+07^ (1.30 × 10^+07^)	nd	nd	nd	Almond and roasted
**6**	Hexanal *^L^*^2^	4.74	44/56/72	800	804	954	1.35 × 10^+10^ (9.21 × 10^+08^)	1.26 × 10^+10^ (9.48 × 10^+08^)	3.11 × 10^+10^ (5.59 × 10^+09^)	1.99 × 10^+10^ (1.90 × 10^+09^)	3.52 × 10^+10^ (3.28 × 10^+09^)	4.33 × 10^+09^ (3.40 × 10^+08^)	nd	6.58 × 10^+09^ (3.98 × 10^+09^)	2.34 × 10^+10^ (3.10 × 10^+09^)	Fatty, green, and grassy
**7**	2-Methyl-2-pentenal *^L^*^2^	5.42	55/69/98	837	931	977	nd	nd	nd	nd	3.54 × 10^+09^ (4.16 × 10^+08^)	nd	nd	nd	nd	Fruity
**8**	(*E*)-2-Hexenal *^L^*^2^	5.98	55/69/83	854	850	941	1.03 × 10^+09^ (5.05 × 10^+07^)	1.42 × 10^+09^ (3.70 × 10^+07^)	6.20 × 10^+09^ (5.38 × 10^+09^)	6.06 × 10^+10^ (3.03 × 10^+09^)	3.21 × 10^+10^ (6.03 × 10^+09^)	1.31 × 10^+10^ (1.40 × 10^+09^)	nd	nd	1.75 × 10^+09^ (1.50 × 10^+09^)	Fresh, fruity, green
**9**	(*Z*)-2-Hexen-1-ol *^L^*^2^	6.30	57/67/82	868	866	909	7.08 × 10^+08^ (1.43 × 10^+08^)	nd	1.27 × 10^+10^ (4.94 × 10^+09^)	nd	nd	nd	nd	nd	nd	Fruity, green, and wine
**10**	(*E*)-4-Hepten-1-al *^L^*^2^	7.19	55/67/84	900	900	936	nd	nd	nd	nd	1.91 × 10^+09^ (1.76 × 10^+09^)	nd	nd	nd	nd	Dairy
**11**	Heptanal *^L^*^2^	7.25	55/70/81	901	902	887	6.79 × 10^+09^ (6.19 × 10^+09^)	1.48 × 10^+09^ (1.44 × 10^+09^)	nd	nd	7.66 × 10^+09^ (1.04 × 10^+10^)	2.15 × 10^+08^ (2.40 × 10^+08^)	nd	9.45 × 10^+09^ (9.77 × 10^+09^)	7.36 × 10^+08^ (1.30 × 10^+09^)	Fresh, green, and citrus
**12**	(*E*,*E*)-2,4-Hexadienal *^L^*^2^	7.48	81	911	910	905	nd	nd	nd	nd	nd	9.29 × 10^+07^ (6,72 × 10^+06^)	nd	nd	nd	
**13**	(*Z*)-2-Heptenal *^L^*^2^	8.84	55/83	958	956	938	1.82 × 10^+10^ (3.50 × 10^+09^)	4.90 × 10^+09^ (9.00 × 10^+08^)	1.24 × 10^+10^ (4.47 × 10^+09^)	3.34 × 10^+09^ (2.44 × 10^+09^)	nd	2.21 × 10^+08^ (3.00 × 10^+07^)	2.00 × 10^+10^ (5.94 × 10^+09^)	nd	1.67 × 10^+09^ (2.90 × 10^+09^)	Almond
**14**	Benzaldehyde *^L^*^1^	8.98	51/77/105	962	961	968	1.65 × 10^+11^ (1.65 × 10^+11^)	5.33 × 10^+10^ (6.94 × 10^+09^)	1.95 × 10^+11^ (9.61 × 10^+10^)	3.29 × 10^+09^ (9.94 × 10^+08^)	1.82 × 10^+10^ (1.04 × 10^+10^)	9.19 × 10^+08^ (1.00 × 10^+08^)	2.48 × 10^+11^ (2.90 × 10^+10^)	4.28 × 10^+09^ (2.59 × 10^+09^)	5.37 × 10^+11^ (2.30 × 10^+11^)	Almond and cherry
**15**	Octanal *^L^*^1^	10.22	57/69	1003	1003	946	4.20 × 10^+09^ (1.09 × 10^+09^)	2.00 × 10^+09^ (2.35 × 10^+08^)	2.54 × 10^+09^ (5.52 × 10^+08^)	nd	1.06 × 10^+09^ (6.07 × 10^+08^)	9.09 × 10^+07^ (3.70 × 10^+07^)	1.56 × 10^+09^ (1.36 × 10^+09^)	1.76 × 10^+09^ (1.41 × 10^+09^)	2.06 × 10^+09^ (2.30 × 10^+08^)	Citrus, fatty, fruit, green, lemon, and honey
**16**	Phenylacetaldehyde *^L^*^1^	11.39	65/91	1045	1042	964	nd	nd	nd	nd	nd	nd	nd	1.56 × 10^+09^ (9.54 × 10^+08^)	nd	Berry, geranium, honey, nut, and pungent
**17**	Benzeneacetaldehyde	11.40	65/91	1045	1042	940	nd	nd	nd	nd	2.57 × 10^+09^ (4.94 × 10^+08^)	nd	nd	nd	nd	Berry, geranium, honey, nut, and pungent
**18**	4-Methyl-benzaldehyde *^L^*^2^	12.58	91/119	1079	1081	940	3.34 × 10^+11^ (5.62 × 10^+11^)	nd	nd	nd	nd	nd	nd	nd	nd	Flavoring agents
**19**	Nonanal*^L^*^1^	13.26	55/77	1104	1104	913	1.28 × 10^+10^ (5.36 × 10^+09^)	9.28 × 10^+09^ (9.45 × 10^+08^)	1.16 × 10^+10^ (1.27 × 10^+09^)	8.29 × 10^+09^ (1.40 × 10^+09^)	1.16 × 10^+10^ (6.68 × 10^+09^)	7.13 × 10^+08^ (3.20 × 10^+08^)	nd	1.08 × 10^+10^ (8.65 × 10^+09^)	nd	Apple, citrus, fruity, grape, green, orange, and rose
**20**	Lilac aldehyde (isomer A) *^L^*^2^	14.32	43/55/93	1145	1140	955	nd	nd	nd	nd	nd	nd	2.04 × 10^+09^ (3.04 × 10^+09^)	2.72 × 10^+10^ (2.12 × 10^+10^)	4.82 × 10^+10^ (5.20 × 10^+09^)	
**21**	Lilac aldehyde (isomer B) *^L^*^2^	14.56	43/55/67/93	1154	1148	946	1.66 × 10^+09^ (2.28 × 10^+08^)	nd	nd	nd	nd	nd	1.35 × 10^+10^ (8.63 × 10^+09^)	6.06 × 10^+10^ (3.77 × 10^+10^)	3.96 × 10^+10^ (3.80 × 10^+10^)	
**22**	Cucumber aldehyde *^L^*^2^	14.64	70	1155	1151	894	nd	nd	nd	nd	nd	4.27 × 10^+07^ (7.02 × 10^+06^)	nd	nd	nd	Cucumber, green, and wax
**23**	(*E*,*Z*)-2,6-nonadienal *^L^*^2^	14.65	70	1155	1153	905	3.00 × 10^+08^ (3.60 × 10^+07^)	nd	nd	nd	nd	nd	nd	nd	nd	Cucumber, green, and wax
**24**	(*E*)-2-Nonenal *^L^*^2^	14.87	55/70/83	1162	1159	893	2.60 × 10^+09^ (5.25 × 10^+08^)	nd	nd	8.84 × 10^+08^ (1.32 × 10^+08^)	nd	nd	nd	nd	nd	Cucumber and green
**25**	Lilac aldehyde (isomer C) *^L^*^2^	14.99	55/93	1169	1163	945	nd	nd	nd	nd	nd	nd	7.66 × 10^+09^ (1.43 × 10^+09^)	2.47 × 10^+10^ (1.53 × 10^+10^)	2.12 × 10^+10^ (3.50 × 10^+09^)	
**26**	Safranal *^L^*^2^	16.01	91/107/121	1201	1198	933	nd	nd	nd	nd	7.84 × 10^+09^ (5.31 × 10^+08^)	nd	nd	nd	nd	Herb
**27**	Decanal *^L^*^1^	16.21	47/67	1206	1205	875	3.91 × 10^+09^ (6.50 × 10^+08^)	9.22 × 10^+09^ (5.89 × 10^+08^)	3.42 × 10^+09^ (3.13 × 10^+08^)	4.42 × 10^+09^ (7.10 × 10^+08^)	nd	1.47 × 10^+08^ (8.10 × 10^+07^)	nd	nd	3.33 × 10^+09^ (4.00 × 10^+08^)	Floral, fried, orange peel, penetrating, and tallow
**28**	2,5-Dimethylbenzaldehyde *^L^*^2^	16.47	77/105/133	1208	1214	928	8.24 × 10^+08^ (1.28 × 10^+08^)	3.90 × 10^+09^ (2.44 × 10^+08^)	nd	nd	nd	nd	nd	nd	nd	
**29**	(*Z*)-2-Decenal *^L^*^2^	17.78	55/70/83	1252	1261	943	2.29 × 10^+09^ (5.55 × 10^+08^)	9.45 × 10^+08^ (7.20 × 10^+07^)	nd	nd	nd	nd	nd	nd	nd	Orange and tallow
**30**	Citral *^L^*^2^	17.93	69	1276	1166	933	nd	nd	nd	nd	1.11 × 10^+09^ (3.45 × 10^+08^)	nd	nd	nd	nd	Lemon
**31**	*p*-Propylbenzaldehyde *^L^*^2^	18.12	91/119/148	-	1273	921	5.69 × 10^+08^ (1.75 × 10^+08^)	1.39 × 10^+09^ (2.88 × 10^+08^)	2.87 × 10^+09^ (1.78 × 10^+08^)	nd	5.22 × 10^+09^ (1.16 × 10^+09^)	2.43 × 10^+08^ (3.70 × 10^+07^)	nd	nd	2.00 × 10^+09^ (2.00 × 10^+08^)	
**Total aldehydes**							**5.69 × 10^+11^**	**1.00 × 10^+11^**	**2.80 × 10^+11^**	**9.94 × 10^+10^**	**1.28 × 10^+11^**	**2.02 × 10^+10^**	**3.12 × 10^+11^**	**1.54 × 10^+11^**	**6.63 × 10^+11^**	
**Alcohols**																
**32**	1-Penten-3-ol *^L^*^2^	2.83	57	684	-	882	nd	nd	nd	nd	1.62 × 10^+09^ (1.67 × 10^+08^)	nd	nd	nd	nd	Butter, fish, green, oxidized, and wet earth
**33**	3-Methyl-1-butanol *^L^*^1^	3.55	55/70	736	-	923	nd	nd	nd	nd	nd	nd	8.92 × 10^+07^ (1.29 × 10^+08^)	1.87 × 10^+09^ (1.17 × 10^+09^)	nd	Burnt, cocoa, floral, and malt
**34**	2-Methyl-1-butanol *^L^*^2^	3.61	43/73	-	-	941	nd	nd	nd	nd	nd	nd	nd	2.89 × 10^+08^ (1.87 × 10^+08^)	nd	Fish oil, green, malt, onion, and wine
**35**	(*Z*)-2-Penten-1-ol *^L^*^2^	4.14	57/68	767	-	917	nd	nd	nd	nd	7.20 × 10^+08^ (1.41 × 10^+08^)	3.84 × 10^+07^ (9.49 × 10^+06^)	nd	nd	nd	Jasmin, green, plastic, and rubber
**36**	3-Hexen-1-ol *^L^*^1^	6.03	55/67/82	857	855	926	nd	1.55 × 10^+09^ (1.72 × 10^+09^)	nd	nd	2.41 × 10^+09^ (9.33 × 10^+08^)	nd	nd	nd	nd	Green
**37**	(*Z*)-2-Hexen-1-ol *^L^*^2^	6.30	57/67	868	866	863	nd	nd	nd	nd	1.18 × 10^+09^ (6.78 × 10^+07^)	nd	nd	nd	nd	
**38**	2-Hexen-1-ol *^L^*^2^	6.31	57/67	862	866	963	nd	8.37 × 10^+09^ (6.53 × 10^+09^)	nd	nd	nd	nd	nd	nd	nd	Green
**39**	1-Hexanol *^L^*^1^	6.40	56/69	868	870	884	nd	1.30 × 10^+10^ (1.60 × 10^+09^)	nd	nd	1.85 × 10^+10^ (3.77 × 10^+09^)	1.40 × 10^+09^ (1.20 × 10^+09^)	2.88 × 10^+09^ (3.92 × 10^+08^)	nd	nd	Banana, flower, grass, and herb
**40**	1-Octen-3-ol *^L^*^2^	9.54	43/55/57	980	980	882	7.02 × 10^+09^ (1.42 × 10^+09^)	1.40 × 10^+10^ (2.60 × 10^+09^)	9.36 × 10^+09^ (9.26 × 10^+08^)	nd	2.87 × 10^+09^ (1.85 × 10^+08^)	1.07 × 10^+08^ (9.30 × 10^+07^)	7.89 × 10^+09^ (2.10 × 10^+09^)	1.51 × 10^+09^ (9.34 × 10^+08^)	7.34 × 10^+09^ (1.20 × 10^+09^)	Fruity, woody, green, mushroom, and nut
**41**	2-Ethyl-1-hexanol *^L^*^2^	10.99	55/57	1030	1029	916	1.48 × 10^+09^ (1.29 × 10^+09^)	1.60 × 10^+10^ (1.67 × 10^+09^)	nd	nd	4.60 × 10^+10^ (5.27 × 10^+10^)	9.96 × 10^+08^ (8.70 × 10^+07^)	7.97 × 10^+09^ (3.05 × 10^+08^)	4.35 × 10^+09^ (2.70 × 10^+09^	8.67 × 10^+09^ (1.10 × 10^+09^)	Oily, rose, and sweet
**42**	Benzyl alcohol *^L^*^2^	11.11	51/79/108	1036	1033	908	3.17 × 10^+09^ (5.57 × 10^+08^)	2.37 × 10^+09^ (2.08 × 10^+09^)	9.07 × 10^+08^ (1.27 × 10^+09^)	nd	nd	nd	nd	nd	nd	Berry, citrus, cherry, floral, and grape
**43**	Phenylethyl alcohol *^L^*^1^	13.45	91	1116	1112	906	7.19 × 10^+08^ (1.17 × 10^+08^)	7.21 × 10^+08^ (6.27 × 10^+08^)	5.52 × 10^+08^ (4.97 × 10^+08^)	nd	nd	8.44 × 10^+06^ (1.50 × 10^+07^)	1.13 × 10^+10^ (3.84 × 10^+08^)	5.53 × 10^+09^ (3.47 × 10^+09^)	3.70 × 10^+09^ (6.00 × 10^+08^)	Fruit, honey, lilac, rose, and wine
**44**	Lilac alcohol (isomer D) *^L^*^2^	16.04	55/93/111	1232	1199	950	nd	nd	nd	nd	nd	nd	nd	nd	nd	
**45**	Lilac alcohol (isomer C) *^L^*^2^	16.42	43/55	1219	1212	953	nd	nd	nd	nd	nd	nd	2.03 × 10^+10^ (1.76E09)	1.90 × 10^+11^ (1.32 × 10^+11^)	nd	
**Total alcohols**						**3.53 × 10^+10^**	**3.31 × 10^+10^**	**1.08 × 10^+10^**	**0.00 × 10^+00^**	**7.33 × 10^+10^**	**2.55 × 10^+09^**	**7.07 × 10^+10^**	**2.04 × 10^+11^**	**1.97 × 10^+10^**		
**Ketones**																
**46**	1-Penten-3-one *^L^*^2^	2.85	55	681	-	822	nd	nd	nd	nd	nd	2.79 × 10^+08^ (2.00 × 10^+07^)	2.51 × 10^+08^ (2.18 × 10^+08^)	nd	nd	Fish, green, mustard, and pungent
**47**	2-Pentanone *^L^*^2^	2.88	43/86	685	-	881	nd	nd	nd	nd	nd	nd	nd	1.94 × 10^+08^ (1.25 × 10^+08^)	nd	Fruit and pungent
**48**	3-Penten-2-one *^L^*^2^	3.59	69/84	733	-	930	nd	nd	6.38 × 10^+07^ (1.11 × 10^+08^)	nd	nd	nd	nd	nd	nd	
**49**	2-Heptanone *^L^*^1^	6.91	43/58	891	890	887	nd	1.66 × 10^+09^ (2.70 × 10^+08^)	3.67 × 10^+08^ (3.19 × 10^+08^)	nd	9.37 × 10^+08^ (1.04 × 10^+08^)	6.05 × 10^+07^ (1.80 × 10^+06^)	3.99 × 10^+08^ (8.94 × 10^+07^)	8.71 × 10^+08^ (5.42 × 10^+08^)	7.35 × 10^+08^ (9.70 × 10^+07^)	Blue cheese, fruit, green, nut, and spice
**50**	4-Methyl-2-heptanone *^L^*^2^	8.25	43/58/85	943	936	923	2.20 × 10^+08^ (3.01 × 10^+07^)	nd	nd	nd	nd	nd	4.99 × 10^+07^ (8.64 × 10^+07^)	9.67 × 10^+07^ (1.19 × 10^+08^)	nd	
**51**	6-Methyl-2-heptanone *^L^*^2^	8.74	43/58	956	953	804	nd	nd	nd	nd	nd	6.22 × 10^+07^ (2.34 × 10^+06^)	nd	2.98 × 10^+08^ (1.81 × 10^+08^)	nd	
**52**	1-Octen-3-one *^L^*^2^	9.43	43/55/70	979	976	905	2.97 × 10^+10^ (4.05 × 10^+10^)	nd	4.62 × 10^+09^ (4.59 × 10^+09^)	nd	nd	nd	2.09 × 10^+10^ (4.68 × 10^+09^)	nd	nd	Mushroom
**53**	6-Methyl-5-heptene-2-one *^L^*^2^	9.66	43/58	986	984	912	nd	8.72 × 10^+09^ (1.40 × 10^+10^)	nd	5.01 × 10^+10^ (4.81 × 10^+09^)	9.20 × 10^+10^ (7.45 × 10^+09^)	9.50 × 10^+09^ (1.20 × 10^+09^)	nd	nd	nd	Citrus, mushroom, pepper, rubber, and strawberry
**54**	1,1,3-Trimethyl-2-cyclohexanone *^L^*^2^	11.15	51/55/82	1036	1034	851	nd	nd	nd	1.85 × 10^+09^ (5.48 × 10^+07^)	1.48 × 10^+09^ (2.58 × 10^+08^)	nd	nd	nd	nd	
**55**	(*E*)-3-Octen-2-one *^L^*^2^	11.24	43/55/111	1033	1035	927	nd	3.56 × 10^+09^ (4.88 × 10^+08^)	3.40 × 10^+09^ (2.95 × 10^+09^)	nd	nd	nd	nd	nd	2.18 × 10^+09^ (2.40 × 10^+08^)	Dull, green, nut, and rose
**56**	3,5,5-Trimethylcyclohex-2-en-1-one *^L^*^2^	11.87	54/82	1124	1058	869	nd	nd	nd	nd	1.81 × 10^+09^ (3.14 × 10^+09^)	nd	nd	nd	1.34 × 10^+09^ (1.20 × 10^+09^)	Cedarwood and spice
**57**	Acetophenone *^L^*^2^	12.05	77/105	1065	1062	946	3.30 × 10^+08^ (4.53 × 10^+07^)	3.01 × 10^+10^ (3.59 × 10^+09^)	1.53 × 10^+09^ (4.17 × 10^+08^)	nd	nd	nd	nd	1.29 × 10^+09^ (8.87 × 10^+08^)	nd	Almonds, flower, meat and must
**58**	2-Nonanone *^L^*^1^	12.83	71	-	1092	1090	nd	nd	nd	nd	nd	nd	nd	2.89 × 10^+08^ (1.77 × 10^+08^)	nd	
**59**	6-Methyl-3,5-heptadiene-2-one *^L^*^2^	13.18	43/79/109	1107	1101	933	nd	nd	nd	nd	1.43 × 10^+09^ (2.80 × 10^+08^)	nd	nd	nd	nd	Spice
**60**	Jasmone *^L^*^2^	21.28	43/79/91/104	1394	1391	798	nd	1.03 × 10^+09^ (8.93 × 10^+08^)	nd	nd	nd	nd	nd	nd	nd	
**61**	Nerylacetone *^L^*^2^	22.66	43/69	1453	1446	915	nd	nd	3.06 × 10^+08^ (5.30 × 10^+08^)	3.48 × 10^+09^ (3.20 × 10^+09^)	1.17 × 10^+10^ (1.81 × 10^+09^)	nd	nd	nd	nd	Fruit
**Total ketones**						**3.03 × 10^+10^**	**4.51 × 10^+10^**	**1.03 × 10^+10^**	**5.54 × 10^+10^**	**1.08 × 10^+11^**	**9.90 × 10^+09^**	**2.16 × 10^+10^**	**3.04 × 10^+09^**	**7.35 × 10^+08^**		
**Esters**																
**62**	Ethyl acetate *^L^*^2^	2.20	43/45/61/70	612	-	933	8.57 × 10^+09^ (3.63 × 10^+09^)	1.59 × 10^+09^ (1.92 × 10^+09^)	nd	1.53 × 10^+09^ (3.22 × 10^+08^)	nd	nd	1.74 × 10^+10^ (2.94 × 10^+09^)	nd	nd	Fruity, pineapple, and pleasant
**63**	Ethyl propanoate *^L^*^2^	3.17	57/77	709	-	828	nd	nd	nd	6.02 × 10^+09^ (9.80 × 10^+09^)	nd	nd	4.16 × 10^+09^ (3.40 × 10^+09^)	nd	nd	Apple, pineapple, rum, and strawberry
**64**	Ethyl isobutyrate *^L^*^1^	3.93	43/71	755	-	924	nd	nd	nd	nd	nd	nd	1.43 × 10^+09^ (1.77 × 10^+08^)	nd	nd	
**65**	(*Z*)-Ethyl crotonate *^L^*^2^	5.77	69/99	830	845	871	nd	nd	nd	nd	nd	nd	7.32 × 10^+08^ (5.37 × 10^+07^)	nd	nd	Tropical fruit
**66**	Ethyl-2-methylbutanoate *^L^*^1^	5.89	57/102	849	850	968	nd	nd	nd	nd	nd	nd	4.72 × 10^+09^ (4.52 × 10^+08^)	nd	nd	
**67**	Ethyl isovalerate *^L^*^1^	6.00	57/88	854	854	892	nd	nd	nd	nd	nd	nd	8.28 × 10^+09^ (4.71 × 10^+08^)	nd	nd	Apple, fruit, pineapple, and sour
**68**	Isoamyl acetate *^L^*^1^	6.59	43/55/70	876	838	880	nd	nd	nd	nd	nd	nd	nd	1.91 × 10^+08^ (2.34 × 10^+08^)	nd	Apple, banana, glue, and pear
**69**	2-Methylbutyl acetate *^L^*^2^	6.65	43/70	880	879	903	nd	nd	nd	nd	nd	nd		3.55 × 10^+08^ (2.15 × 10^+08^)	nd	Ester, fresh, fruit, and pineapple
**70**	Ethyl pentanoate *^L^*^2^	7.24	57/85/133	900	902	897	nd	nd	nd	nd	nd	nd	1.32 × 10^+10^ (1.10 × 10^+10^)	nd	nd	Apple, dry fish, herb, nut, and yeast
**71**	Amyl acetate *^L^*^2^	7.60	43/55/70	828	914	828	nd	nd	nd	nd	nd	nd	2.76 × 10^+08^ (9.83 × 10^+07^)	nd	nd	Banana
**72**	Methyl hexanoate *^L^*^2^	7.89	43/74/87	925	924	846	nd	nd	1.88 × 10^+08^ (1.65 × 10^+08^)	nd	nd	2.93 × 10^+07^ (2.80 × 10^+07^)	nd	5.62 × 10^+08^ (3.40 × 10^+08^)	8.27 × 10^+08^ (7.11 × 10^+06^)	Ester, fresh, fruit, and pineapple
**73**	Methyl hexanoate *^L^*^2^	7.90	43/59/74	925	924	874	nd	3.12 × 10^+08^ (3.20 × 10^+08^)	nd	nd	nd	nd	nd	nd	nd	Ester, fresh, fruit, and pineapple
**74**	Ethyl (*E*)-2-pentenoate *^L^*^2^	8.59	55/83/99	-	948	908	nd	nd	nd	nd	nd	nd	3.45 × 10^+08^ (5.98 × 10^+08^)	nd	nd	
**75**	Ethyl hexanoate *^L^*^1^	10.10	43/70/88/99	1000	999	911	3.34 × 10^+10^ (8.90 × 10^+08^)	nd	nd	3.54 × 10^+10^ (3.33 × 10^+09^)	nd	1.19 × 10^+08^ (3.30 × 10^+07^)	4.13 × 10^+10^ (1.70 × 10^+08^)	nd	nd	Apple peel, brandy, fruit gum, overripe fruit, and pineapple
**76**	Ethyl 2-hexenoate *^L^*^2^	11.43	55/97	1037	1043	875	nd	nd	nd	nd	nd	nd	4.13 × 10^+09^ (1.70 × 10^+08^)	nd	nd	
**77**	Methyl benzoate *^L^*^2^	12.91	51/77/105/136	1094	1092	886	nd	5.76 × 10^+09^ (2.80 × 10^+08^)	nd	nd	nd	nd	nd	nd	nd	Herb, lettuce, prune, and violet
**78**	Ethyl heptanoate *^L^*^1^	13.04	43/88	1097	1097	921	2.56 × 10^+09^ (2.31 × 10^+08^)	nd	nd	3.38 × 10^+09^ (4.04 × 10^+09^)	nd	nd	1.10 × 10^+09^ (5.18 × 10^+08^)	nd	nd	Brandy, fruit, and wine
**79**	Ethyl benzoate *^L^*^2^	15.17	77/105	1171	1169	933	1.30 × 10^+10^(9.88 × 10^+09^)	nd	1.32 × 10^+09^ (9.99 × 10^+07^)	nd	nd	nd	nd	2.73 × 10^+10^ (3.55 × 10^+09^)	nd	Chamomile, celery, fat, flower, and fruit
**80**	Diethyl succinate *^L^*^1^	15.41	101/129	1182	1177	885	nd	nd	nd	nd	nd	nd	nd	5.93 × 10^+08^ (1.03 × 10^+09^)	nd	Cotton, fabric, floral, fruit, and wine
**81**	Methyl salicylate *^L^*^2^	15.81	92/120/152	1192	1191	910	nd	nd	nd	nd	3.30 × 10^+09^ (8.01 × 10^+08^)	1.32 × 10^+08^ (1.70 × 10^+07^)	3.30 × 10^+09^ (8.01 × 10^+08^)	nd	nd	
**82**	Ethyl octanoate *^L^*^1^	15.97	47/88/101	1196	1197	919	1.51 × 10^+10^ (1.30 × 10^+10^)	nd	nd	1.54 × 10^+09^ (1.78 × 10^+08^)	nd	2.61 × 10^+07^ (9.27 × 10^+06^)	nd	5.58 × 10^+09^ (3.38 × 10^+09^)	nd	Apricot, brandy, fat, floral, and pineapple
**83**	Ethyl phenylacetate *^L^*^1^	17.21	65/91/164	1246	1241	907	nd	nd	nd	nd	nd	nd	nd	9.67 × 10^+09^ (7.77 × 10^+08^)	nd	Floral, fruit, honey, and rose
**84**	Ethyl (*E*)-2-octenoate *^L^*^2^	17.35	55/73/125	1249	1246	903	nd	nd	nd	nd	nd	nd	nd	2.43 × 10^+09^ (1.22 × 10^+08^)	nd	Fruit
**85**	Ethyl nonanoate *^L^*^1^	18.70	88/101	1296	1294	903	4.41 × 10^+09^ (5.27 × 10^+08^)	nd	nd	1.13 × 10^+09^ (2.85 × 10^+08^)	nd	nd	nd	nd	nd	Floral
**86**	Ethyl phenylpropanoate *^L^*^2^	20.10	91/104/178	1353	1347	799	nd	nd	nd	nd	nd	nd	8.73 × 10^+07^ (1.51 × 10^+08^)	nd	nd	Flower and honey
**87**	Ethyl undecanoate *^L^*^2^	23.8	43/88/101	1494	1492	881	nd	nd	nd	nd	nd	nd	3.62 × 10^+08^ (1.49 × 10^+08^)	nd	nd	Coconut and cognac
**88**	Methyl dihydrojasmonate *^L^*^2^	27.42	83	1649	1646	836	1.34 × 10^+08^ (6.74 × 10^+07^)	nd	nd	nd	2.10 × 10^+08^ (4.74 × 10^+06^)	nd	nd	nd	nd	Floral and jasmine
**89**	Ethyl myristate *^L^*^2^	30.60	88/101	1794	1793	882	nd	nd	nd	nd	nd	nd	8.04 × 10^+08^ (2.12 × 10^+08^)	nd	nd	Wax
**90**	Isopropyl myristate *^L^*^2^	31.21	43/60/228	1827	1822	873	1.73 × 10^+08^ (5.13 × 10^+07^)	2.49 × 10^+08^ (7.47 × 10^+06^)	1.32 × 10^+08^ (2.67 × 10^+07^)	1.20 × 10^+08^ (4.77 × 10^+07^)	2.96 × 10^+08^ (1.14 × 10^+08^)	1.22 × 10^+07^ (1.18 × 10^+06^)	1.15 × 10^+08^ (2.24 × 10^+07^)	1.09 × 10^+08^ (6.73 × 10^+07^)	1.28 × 10^+08^ (6.60 × 10^+07^)	Dairy
**91**	Ethyl palmitate *^L^*^2^	34.56	88/101	1993	1993	906	nd	nd	nd	nd	nd	nd	4.19 × 10^+09^ (1.18 × 10^+09^)	nd	nd	Wax
**Total esters**						**7.73 × 10^+10^**	**7.91 × 10^+09^**	**1.64 × 10^+09^**	**4.91 × 10^+10^**	**3.81 × 10^+09^**	**3.19 × 10^+08^**	**1.01 × 10^+11^**	**4.68 × 10^+10^**	**9.55 × 10^+08^**		
**Acids**																
**92**	Acetic acid *^L^*^2^	3.14	43/45/60	610	-	926	nd	8.95 × 10^+08^ (7.76 × 10^+08^)	1.39 × 10^+09^ (4.33 × 10^+08^)	nd	3.19 × 10^+08^ (2.80 × 10^+08^)	nd	nd	1.33 × 10^+09^ (8.56 × 10^+08^)	4.88 × 10^+09^ (1.90 × 10^+09^)	Acid, fruit, pungent, sour, and vinegar
**93**	Octanoic acid *^L^*^2^	15.30	43/55/60/73	1180	1174	869	nd	nd	nd	nd	nd	nd	2.53 × 10^+09^ (1.98 × 10^+09^)	nd	nd	Cheese, fat, grass, and oil
**94**	Nonanoic acid *^L^*^2^	17.97	69/73/92/120	1273	1268	913	8.91 × 10^+08^ (1.48 × 10^+09^)	6.33 × 10^+08^ (6.58 × 10^+08^)	3.81 × 10^+08^ (4.74 × 10^+09^)	1.06 × 10^+09^ (5.34 × 10^+08^)	nd	9.53 × 10^+07^ (9.90 × 10^+07^)			4.03 × 10^+09^ (1.10 × 10^+09^)	Fat, green, and sour
**95**	Decanoic acid *^L^*^2^	30.55	60/73/129	nd	nd	nd	nd	nd	nd	nd	nd	nd	nd	4.29 × 10^+09^ (5.26 × 10^+09^)	nd	Dust, fat, and grass
**Total acids**						**8.91 × 10^+08^**	**1.53 × 10^+09^**	**1.77 × 10^+09^**	**1.06 × 10^+09^**	**3.19 × 10^+08^**	**9.53 × 10^+07^**	**2.53 × 10^+09^**	**5.62 × 10^+09^**	**8.91 × 10^+09^**		
**Monoterpenes**																
**96**	Limonene *^L^*^1^	11.01	67/79/93	1030	1029	904	nd	nd	3.49 × 10^+10^ (3.23 × 10^+09^)	nd	nd	nd	nd	nd	nd	Coriander, floral, lavender, lemon, and rose
**97**	Terpinolene *^L^*^2^	12.71	79/93/121/136	1088	1086	953	nd	nd	1.87 × 10^+09^ (1.70 × 10^+09^)	nd	nd	nd	nd	nd	nd	Pine
**98**	Linalool *^L^*^1^	13.12	55/71/93	1099	1099	855	nd	nd	nd	nd	2.98 × 10^+09^ (4.99 × 10^+08^)	nd	nd	5.58 × 10^+09^ (3.49 × 10^+09^)	nd	
**99**	α-Terpineol *^L^*^2^	15.91	59/93/121	1189	1194	926	nd	4.49 × 10^+09^ (2.56 × 10^+08^)	nd	nd	nd	nd	nd	nd	nd	Anise, fresh, mint, and oil
**Total monoterpenes**					**0.00 × 10^+00^**	**4.49 × 10^+09^**	**3.68 × 10^+10^**	**0.00 × 10^+00^**	**2.98 × 10^+09^**	**0.00 × 10^+00^**	**0.00 × 10^+00^**	**5.58 × 10^+09^**	**0.00 × 10^+00^**			
**Norisoprenoids**																
**100**	β-Cyclocitral *^L^*^1^	16.59	67/81/109/137/152	1220	1218	940	nd	nd	nd	1.90 × 10^+09^ (2.76 × 10^+08^)	1.45 × 10^+10^ (1.13 × 10^+09^)	5.81 × 10^+08^ (5.40 × 10^+07^)	nd	nd	nd	
**101**	α-Ionone *^L^*^1^	22.04	43/93	1426	1421	795	nd	2.49 × 10^+08^ (7.47^+06^)	nd	5.00 × 10^+08^ (1.26 × 10^+08^)	2.03 × 10^+09^ (1.32 × 10^+08^)	5.60 × 10^+07^ (2.81^+06^)	nd	nd	nd	Floral and violet
**102**	β-Ionone *^L^*^2^	23.46	43/93/177	1491	1478	878	nd	nd	nd	nd	5.53 × 10^+09^ (2.57 × 10^+08^)	2.61 × 10^+08^ (1.40 × 10^+07^)	nd	nd	nd	Floral and violet
**Total norisoprenoids**					**0.00 × 10^+00^**	**2.49 × 10^+08^**	**0.00 × 10^+00^**	**2.40 × 10^+09^**	**2.21 × 10^+10^**	**8.98 × 10^+08^**	**0.00 × 10^+00^**	**0.00 × 10^+00^**	**0.00 × 10^+00^**			
**Hydrocarbons**																
**103**	2,4-Dimethyl-1-heptene *^L^*^2^	5.66	43/55/70	836	841	965	nd	nd	2.62 × 10^+09^ (3.45 × 10^+08^)	nd	nd	8.68 × 10^+08^ (1.90 × 10^+08^)	3.82 × 10^+09^ (2.96 × 10^+08^)	nd	3.36 × 10^+09^(5.60 × 10^+08^)	
**104**	1,3-Dimethylheptane *^L^*^2^	6.26	43/71/85	863	864	943	nd	nd	nd	nd	nd	nd	1.27 × 10^+09^ (3.75 × 10^+07^)	nd	nd	
**105**	Styrene *^L^*^2^	6.98	51/78/104	893	892	937	2.80 × 10^+09^ (3.59 × 10^+08^)	2.47 × 10^+09^ (2.95 × 10^+08^)	4.25 × 10^+09^ (5.50 × 10^+08^)	nd	nd	4.56 × 10^+07^ (4.30 × 10^+07^)	8.77 × 10^+09^ (6.85 × 10^+09^)	nd	nd	
**106**	*o*-Cymene *^L^*^2^	10.87	91/119	1022	1025	955	nd	nd	1.26 × 10^+10^ (7.35 × 10^+09^)	nd	nd	nd	nd	nd	nd	
**Total hydrocarbons**						**2.80 × 10^+09^**	**2.47 × 10^+09^**	**1.95 × 10^+10^**	**0.00 × 10^+00^**	**0.00 × 10^+00^**	**9.14 × 10^+08^**	**1.39 × 10^+10^**	**0.00 × 10^+00^**	**3.36 × 10^+09^**		
**Heterocyclics**																
**107**	2-Methylfuran *^L^*^2^	2.19	43/53/82	606	-	826	nd	nd	nd	1.00 × 10^+09^ (1.68 × 10^+08^)	4.18 × 10^+08^ (3.60 × 10^+07^)	nd	nd	nd	nd	
**108**	Methyltetrahydrofuran *^L^*^2^	2.70	43/71	674	-	876	nd	nd	nd		2.86 × 10^+08^ (5.87 × 10^+07^)	nd	nd	nd	nd	
**109**	2-Ethylfuran *^L^*^2^	3.03	53/81	703	-	960	nd	nd	nd	3.94 × 10^+09^ (4.06 × 10^+09^)	1.42 × 10^+10^ (3.72 × 10^+09^)	1.27 × 10^+09^ (1.30 × 10^+09^)	nd	nd	nd	Butter and caramel
**110**	2,4-Dimethylfuran *^L^*^2^	3.22	53/67/96	729	-	850	nd	nd	nd	nd	nd	nd	nd	2.77 × 10^+08^ (1.85 × 10^+08^)	nd	
**110**	2,4-Dimethylfuran *^L^*^2^	3.22	53/67/96	729	-	850	nd	nd	nd	nd	nd	nd	nd	2.77 × 10^+08^ (1.85 × 10^+08^)	nd	
**111**	Furfural *^L^*^1^	5.44	95	833	832	878	nd	nd	nd	nd	nd	nd	nd	2.30 × 10^+09^ (1.39 × 10^+09^)	2.50 × 10^+09^ (7.00 × 10^+07^)	Almond, baked potatoes, bread, burnt, and spice
**112**	2,3-Dimethylpyrazine *^L^*^2^	7.66	67/108	925	916	905	nd	nd	nd	nd	nd	nd	nd	3.80 × 10^+08^ (2.49 × 10^+08^)	nd	Caramel, cocoa, hazelnut, peanut butter, and roasted
**113**	2-Penthylfuran *^L^*^2^	9.84	81	993	990	924	nd	nd	nd	1.70 × 10^+09^ (1.48 × 10^+09^)	nd	nd	nd	nd	nd	Butter, floral, fruit, and green bean
**Total heterocyclics**						**0.00 × 10^+00^**	**0.00 × 10^+00^**	**0.00 × 10^+00^**	**6.64 × 10^+09^**	**1.49 × 10^+10^**	**1.27 × 10^+09^**	**0.00 × 10^+00^**	**2.96 × 10^+09^**	**2.50 × 10^+09^**		
**Lactones**																
**114**	4-Methyl-4-vinylbutyrolactone *^L^*^2^	11.17	55/67/111	1043	1035	923	nd	nd	nd	nd	nd	nd	6.54 × 10^+09^ (7.67 × 10^+08^)	5.74 × 10^+09^ (3.60 × 10^+09^)	nd	
**Total lactones**						**0.00 × 10^+00^**	**0.00 × 10^+00^**	**0.00 × 10^+00^**	**0.00 × 10^+00^**	**0.00 × 10^+00^**	**0.00 × 10^+00^**	**6.54 × 10^+09^**	**5.74 × 10^+09^**	**0.00 × 10^+00^**		
**Phenols**																
**115**	4-Ethylphenol *^L^*^2^	15.16	107/122	1169	1169	921	nd	nd	nd	nd	nd	nd	nd	4.43 × 10^+09^ (3.09 × 10^+09^)	nd	Leather, phenol, spice, and stable
**Total phenols**						**0.00 × 10^+00^**	**0.00 × 10^+00^**	**0.00 × 10^+00^**	**0.00 × 10^+00^**	**0.00 × 10^+00^**	**0.00 × 10^+00^**	**0.00 × 10^+00^**	**4.43 × 10^+09^**	**0.00 × 10^+00^**		
**Phenylpropenes**																
**116**	Estragole *^L^*^2^	15.99	51/77/91/121/133/148	1196	1197	883	nd	nd	nd	nd	nd	6.45 × 10^+07^ (5.70 × 10^+07^)	nd	nd	nd	Anise and licorice
**117**	Apioline *^L^*^2^	27.99	149/222	1682	1672	865	nd	nd	nd	nd	nd	5.12 × 10^+06^ (8.88 × 10^+06^)	nd	nd	nd	
**Total phenylpropenes**						**0.00 × 10^+00^**	**0.00 × 10^+00^**	**0.00 × 10^+00^**	**0.00 × 10^+00^**	**0.00 × 10^+00^**	**6.96 × 10^+07^**	**0.00 × 10^+00^**	**0.00 × 10^+00^**	**0.00 × 10^+00^**		

RT = retention time (minutes). ^a^ retention index reported in literature. ^b^ retention index determined using a commercial hydrocarbon mixture (C6–C20). Rmatch: identification method tentatively identified by the NIST14 Library Database. ^c^ Area expressed as arbitrary units; (S.D. = standard deviation of three independent assays). *^L^*^1^ Identified metabolites (identification according to the comparison of the RT and MS fragmentation of a commercial reference standard and the analysis was performed under identical analytical conditions) [59]. *^L^*^2^ Putatively annotated compounds (spectral (MS) similarity with the NIST14 database) [59]. nd = not detectable.

According to Figure 4, it is possible to verify that the richest extracts in heterocyclics, norisoprenoids, ketones, phenylpropenes, and monoterpenes (all leaf extracts, and stem hydroethanolic extract) were mainly found in the positive F1 axis. On the other hand, the extracts that presented higher contents of aldehydes and hydrocarbons but the lowest levels of heterocyclics, norisoprenoids, and ketones were projected along the negative F1 and F2 axes (stem hydroethanolic extract and flower crude extract). Additionally, the extracts richest in alcohols, phenols, lactones, acids, and esters were found on the positive F2 axis (flower aqueous infusion and hydroethanolic extract).

## 4. Conclusions

This work allowed a deeper understanding of some essential and non-essential elements and volatile profiles of the stems, leaves, and flowers of *P. avium*. The obtained data showed that these by-products have essential minerals, with phosphorus being the most abundant essential macromineral in all samples. Regarding the trace elements, stems showed significant levels of Fe, while flowers showed more Mn and Zn. In addition, a total of 117 VOCs were identified in the different sweet cherry by-product extracts. Among all VOCs analyzed, aldehydes, esters, ketones, and alcohols were the main classes found. Benzaldehyde, 4-methyl-benzaldehyde, hexanal, lilac aldehyde, and 6-methyl-5-hepten-2-one were those present in the highest amounts. It should be noted that, as far as we know, this is the first work to determine the mineral and volatile composition of sweet cherry by-products of the Saco cultivar collected in the Fundão region (Portugal). Thus, by complementing the phenolic profiles evaluated in previous studies, we can conclude that the stems, leaves, and flowers of Portuguese *P. avium* may have great potential as food supplements and natural flavoring agents to be used in the pharmaceutical and food industries. However, these by-products should be further explored. The extraction of the most promising bioactive compounds in each matrix or the potential use of beverages based on infusions of sweet cherry stems and leaves to improve human health may be ways to take advantage of these bio-wastes. On the other hand, the use of sweet cherry by-products as antioxidant preservatives for the food industry or their potential against antibiotic resistance may be other possibilities to be studied.

## Figures and Tables

**Figure 1 foods-11-00751-f001:**
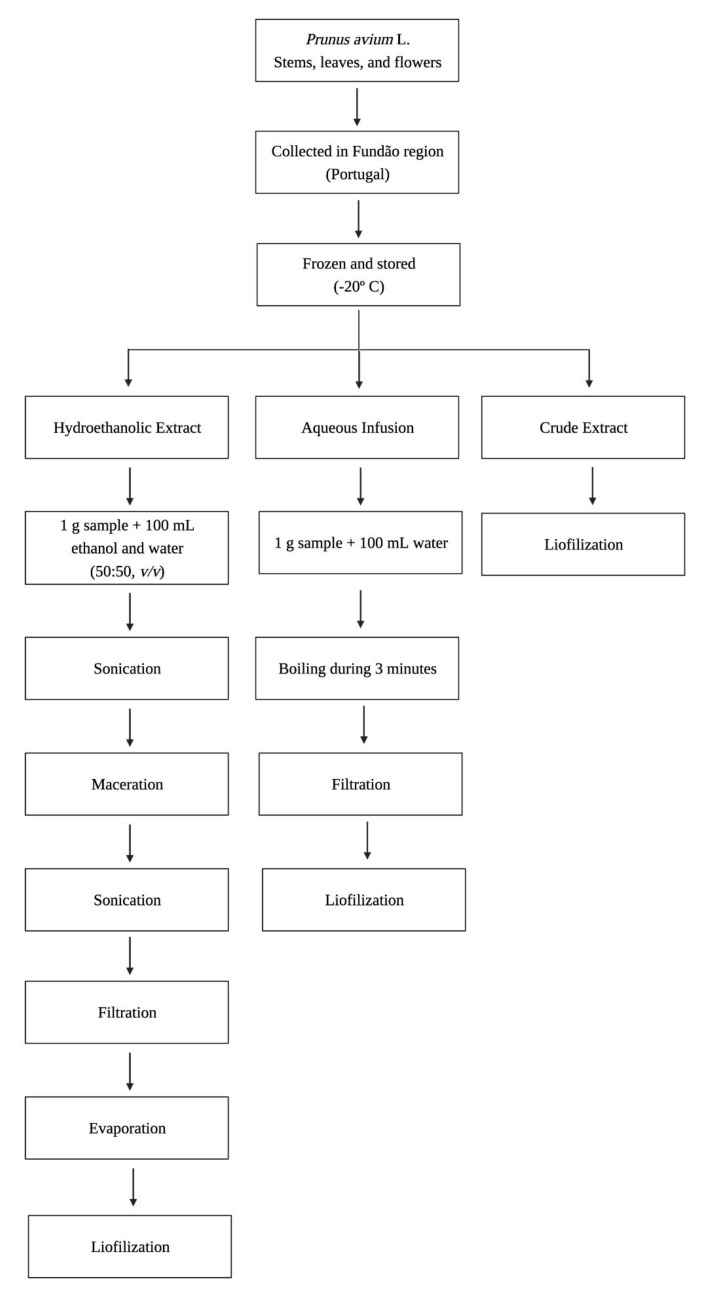
Scheme of *P. avium* L. by-product hydroethanolic extract, aqueous infusion, and crude extract preparation.

**Figure 2 foods-11-00751-f002:**
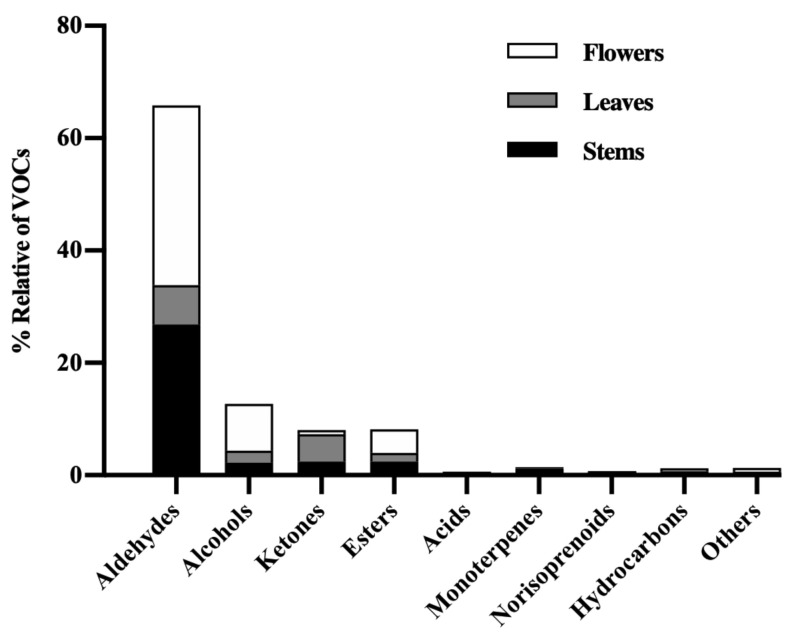
Percentage of the total volatile organic compounds (VOCs) represented by the different classes of compounds in the *Prunus avium* L. stems, leaves, and flowers from the Saco variety.

**Figure 3 foods-11-00751-f003:**
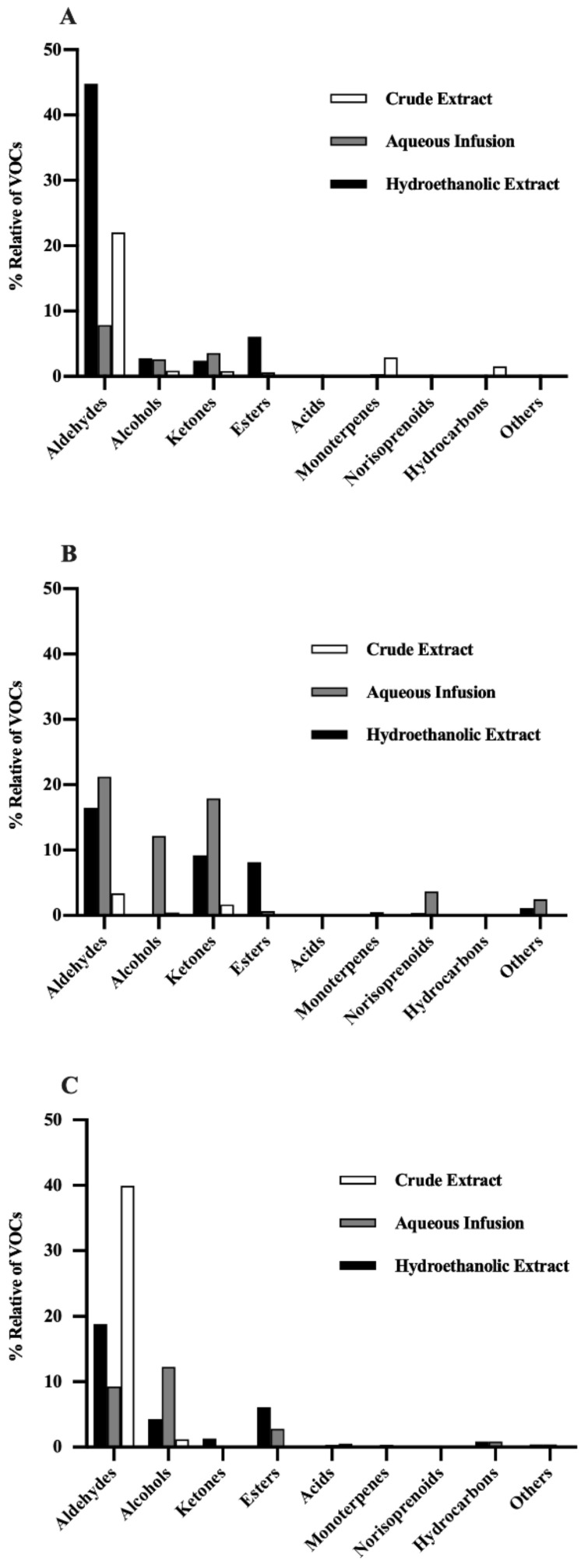
Percentage of the total volatile organic compounds (VOCs) represented by the different classes of compounds in the hydroethanolic extract, aqueous infusion, and crude extract of *Prunus avium* L. (**A**) stems, (**B**) leaves, and (**C**) flowers from the Saco variety.

**Figure 4 foods-11-00751-f004:**
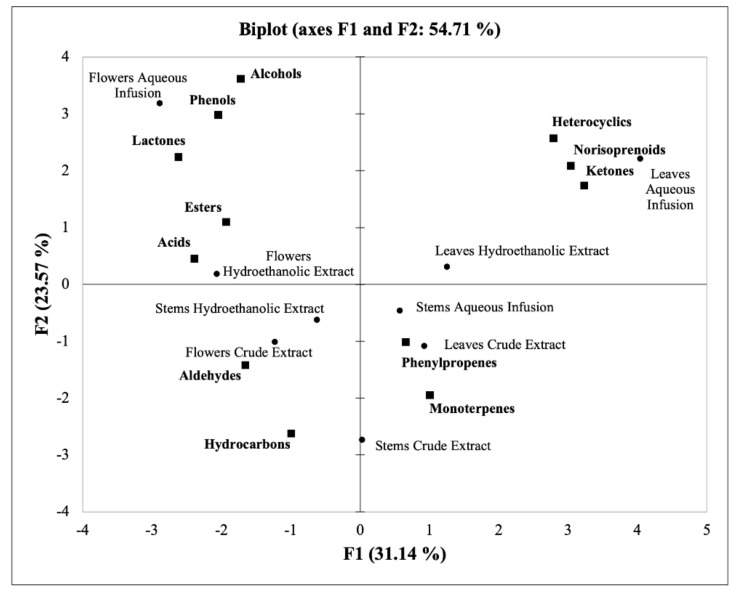
Principal component analysis (PCA) of all classes of organic volatile compounds (VOCs) (aldehydes, alcohols, ketones, esters, acids, monoterpenes, norisoprenoids, hydrocarbons, lactones, phenols, and phenylpropenes) identified in the *Prunus avium* L. stem, leaf, and flower extracts (hydroethanolic, aqueous infusion, and crude) in the plane composed by the principal axes F1 and F2 (54.71%). The samples are represented with a dot, and VOCs classes are represented with a square.

**Table 1 foods-11-00751-t001:** Content of the eight essential elements (mg/kg of dw) found in the *Prunus avium* L. by-products from the Saco variety from the Fundão region (Portugal).

Element	Cherry Stems	Cherry Leaves	Cherry Flowers
** *Macrominerals* **			
**Sodium, Na**	79.5 ± 6.0	<LOQ	<LOQ
**Phosphorus, P**	1345 ± 54 ^b^	1058 ± 112 ^a^	837 ± 10 ^a,b^
** *Trace elements* **			
**Cobalt, Co**	<LOQ	0.057 ± 0.005	<LOQ
**Copper, Cu**	24.8 ± 5.5 ^b,c^	4.22 ± 0.31 ^a^	6.79 ± 0.34 ^a^
**Iron, Fe**	32.46 ± 0.92 ^b^	66.8 ± 4.1 ^a^	30.9 ± 2.5 ^b^
**Manganese, Mn**	10.72 ± 0.39 ^b^	95.0 ± 8.3 ^a^	18.39 ± 0.35 ^b^
**Selenium, Se**	<LOD	<LOD	<LOD
**Zinc, Zn**	18.00 ± 1.3 ^c^	20.9 ± 4.2 ^c^	35.8 ± 2.8 ^a,b^

The values are expressed as means ± standard deviations of three independent determinations. Significant differences between the by-products according to the Tukey’s test (*p* < 0.05) are indicated by: ^a^, vs. cherry stems; ^b^, vs. cherry leaves; and ^c^, vs. cherry flowers.

**Table 2 foods-11-00751-t002:** Content of the 10 non-essential elements (mg/kg of dw) found in the *Prunus avium* L. by-products from the Saco variety from the Fundão region (Portugal).

Element	Cherry Stems	Cherry Leaves	Cherry Flowers
**Aluminum, Al**	7.30 ± 0.54 ^b^	32.2 ± 3.2 ^a,c^	20.9 ± 1.5 ^a,b^
**Arsenic, As**	<LOD	<LOD	<LOD
**Barium, Ba**	42.99 ± 0.59 ^b^	38.9 ± 2.5 ^a,c^	4.26 ± 0.37 ^a,b^
**Cadmium, Cd**	<LOD	<LOD	<LOD
**Chromium, Cr**	<LOQ	<LOQ	<LOQ
**Lithium, Li**	0.07 ± 0.01 ^b^	0.02 ± 0.00 ^a,c^	0.04 ± 0.00 ^a,b^
**Nickel, Ni**	<LOQ	<LOQ	<LOQ
**Lead, Pb**	0.35 ± 0.05 ^b^	0.21 ± 0.02 ^a,c^	0.04 ± 0.00 ^a,b^
**Rubidium, Rb**	7.69 ± 0.16 ^b^	5.34 ± 0.38 ^a,c^	3.88 ± 0.07 ^a,b^
**Strontium, Sr**	18.62 ± 0.53 ^c^	20.3 ± 1.2 ^c^	2.89 ± 0.17 ^a,b^

The values are expressed as means ± standard deviations of three independent determinations. Significant differences between the by-products according to the Tukey’s test (*p* < 0.05) are indicated by: ^a^, vs. cherry stems; ^b^, vs. cherry leaves; and ^c^, vs. cherry flowers.

## Data Availability

The data are contained within this article.

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
