# Peer review of "Mineral Content and Volatile Profiling of Prunus avium L. (Sweet Cherry) By-Products from Fundão Region (Portugal)"

_foods, 2022, doi:10.3390/foods11050751_

Round 1

Reviewer 1 Report

The authors present a work entitled "Mineral Content and Volatile Profiling of Prunus avium L. 2 (Sweet Cherry) By-Products from Fundão Region (Portugal) 3". By-products represent a much studied topic ion the scientific community, so this work could be considered an useful contribution to this field. The manuscript is well written and organized, the findings are presented in correct way using exemplasry tables. The discussion is rich and encloses the recent literature while the seem well organized with a nod to the possible industrial application of these recovered products

Minor issues:

The authors report a high level of K but in the text and tables there is no discussion about it, please add any comments about its concentration and the important role that it plays for humane health.

Author Response

The authors would like to thank the Reviewers for carefully reviewing our manuscript and providing us with their comments and suggestions to improve the quality of the work. The following responses have been prepared to address all the reviewer comments in a point-by-point table template. Additionally, all modifications made in the manuscript addressing the comments of Reviewers are highlighted in yellow.

Reviewer

Comment

Response

1

The authors present a work entitled "Mineral Content and Volatile Profiling of Prunus avium L. (Sweet Cherry) By-Products from Fundão Region (Portugal)". By-products represent a much studied topic ion the scientific community, so this work could be considered an useful contribution to this field. The manuscript is well written and organized, the findings are presented in correct way using exemplary tables. The discussion is rich and encloses the recent literature while the seem well organized with a nod to the possible industrial application of these recovered products.

Minor issues:

The authors report a high level of K but in the text and tables there is no discussion about it, please add any comments about its concentration and the important role that it plays for human health

The authors are grateful for the reviewer’s comment and suggestion. In fact, the authors report a high level of potassium (K) but in red fruits, particularly in cherry fruit. Although this mineral is one of the body's electrolytes and it is essential for the normal functioning of cells, nerves and muscles, its presence was not detected in our experiment. For this reason is that there is no reference or discussion about this element in the tables and text.

Reviewer 2 Report

Paper reports the results of investigations carried out correctly, applying modern equipment and methodology. Results may support further utilization of sweet cherry and its by-products.

I suggest minor revision in Keywords: words in the title should'nt be repeated as key-words (e.g. by-products; volatile compounds; mineral elements;)

Little spell check is required (e.g. in line 91 "have not yet been described")

Author Response

The authors would like to thank the Reviewers for carefully reviewing our manuscript and providing us with their comments and suggestions to improve the quality of the work. The following responses have been prepared to address all the reviewer comments in a point-by-point table template. Additionally, all modifications made in the manuscript addressing the comments of Reviewers are highlighted in yellow.

Reviewer

Comment

Response

2

Paper reports the results of investigations carried out correctly, applying modern equipment and methodology. Results may support further utilization of sweet cherry and its by-products.

I suggest minor revision in Keywords: words in the title should'nt be repeated as key-words (e.g. by-products; volatile compounds; mineral elements;)

Little spell check is required (e.g. in line 91 "have not yet been described")

The authors thank by the reviewer’s comment and suggestions. As suggested, the Keywords were revised. (Please see now lines 40 and 41 of the revised manuscript).

As suggested, minor errors were corrected, and the English was revised. (Please see now the revised version of manuscript).

Reviewer 3 Report

In general, it is an interesting study that presents the potential of the use of cherry by-products, however, more detailed information and more specific conclusions should be provided.

We have to keep in mind that cherry flowers, leaves and stems contain cyanogenic compounds which break down when chewed or crushed to release cyanide. So this topic should be also taken into account and discussed. The analyzed parts of sweet cherry contain amygdalin which is highly toxic or as other studies say that it has an anti-cancer effect. I understand that the Authors did not focus on this in the manuscript however, it would be recommended to take it into account.

It would be also good to describe how the by-products could be used exactly.

As all the values (minerals content) are given per dry matter it would be recommended to provide an average water content of leaves, flowers and stems.

The Authors should check and improve the references list because different styles are used.

Lines 193-194: "The oven temperature was set at 40 °C (for 1 min), then increased from 5 °C/min to 250 °C and held for 5 min” – in my point of view it is a bit misleading, consider replacing it with sth like "increased from 40°C to 250°C with a speed of 5°C /min…".

In the Results the Authors present the total mineral content, however in the Methods it is not clearly explained how it how analyzed, please add it to the Methods section 2.3.

“Macrominerals” (f.ex. line 216) and “macro-minerals” (f.ex. line 229)  decide for one typing in the whole manuscript.  

Line 239: “This element is present in fruits in the range of 9.9–94.3 mg/100 g” is it the amount of P in dry matter or fresh product? If it is per fresh product it would be easier to compare values when they are also expressed per dry matter. It is also not clear which fruits? Sweet cherry?

Conclusions

Line 538: “This work allowed a deeper understanding of the mineral content…” Rewrite this sentence. In fact, the Authors analyzed only some of the minerals.

In my opinion, more precise conclusions are missing. The part (lines 550-552): “may have great potential as food supplements and natural flavoring agents to be used in the pharmaceutical and food industries.” Is very general. More detailed possibilities of using the by-products of sweet cherry flowers, leaves and stems are missing.

Author Response

The authors would like to thank the Reviewers for carefully reviewing our manuscript and providing us with their comments and suggestions to improve the quality of the work. The following responses have been prepared to address all the reviewer comments in a point-by-point table template. Additionally, all modifications made in the manuscript addressing the comments of Reviewers are highlighted in yellow.

Reviewer

Comment

Response

3

In general, it is an interesting study that presents the potential of the use of cherry by-products, however, more detailed information and more specific conclusions should be provided.

We have to keep in mind that cherry flowers, leaves and stems contain cyanogenic compounds which break down when chewed or crushed to release cyanide. So this topic should be also taken into account and discussed. The analyzed parts of sweet cherry contain amygdalin which is highly toxic or as other studies say that it has an anti-cancer effect. I understand that the Authors did not focus on this in the manuscript however, it would be recommended to take it into account.

The authors thank the reviewer’s comment, which contributed to enhance the quality and reading of the manuscript. We followed the reviewer’s suggestions and inserted a brief discussion about cyanogenic compounds and amygdalin in the section 2.3. of the Results and Discussion of the manuscript (Please see now lines 463 to 471 of the revised manuscript).

It would be also good to describe how the by-products could be used exactly.

The authors thank the reviewer’s suggested. We describe how the products can be used in the Conclusion section. (Please see now lines 562 to 567 of the revised manuscript).

As all the values (minerals content) are given per dry matter it would be recommended to provide an average water content of leaves, flowers and stems.

The authors thank the reviewer’s suggestion and they understand your point of view. However, the focus of this work was the content in essential and non-essential elements, and volatile organic compounds. In a future paper, we may consider the reviewer's recommendation and evaluate the average water content of sweet cherry stems, leaves, and flowers.

The Authors should check and improve the references list because different styles are used.

The authors thank the reviewer’s comment. As suggested the references list was revised. (Please see now the References section of revised version of manuscript).

Lines 193-194: "The oven temperature was set at 40 °C (for 1 min), then increased from 5 °C/min to 250 °C and held for 5 min” – in my point of view it is a bit misleading, consider replacing it with sth like "increased from 40°C to 250°C with a speed of 5°C /min…".

As suggested, the lines 193-194 was reformulated. Thank you for the suggestion. (Please see now lines 193 and 194 of the revised manuscript).

In the Results the Authors present the total mineral content, however in the Methods it is not clearly explained how it how analyzed, please add it to the Methods section 2.3.

The authors thank the reviewer’s comment and suggestion. The total mineral content was done by summing the content of each mineral.

“Macrominerals” (f.ex. line 216) and “macro-minerals” (f.ex. line 229) decide for one typing in the whole manuscript.

The authors thank the reviewer’s suggested. The word “macrominerals” was adopted. (Please see now lines 215, 228, 261, and 551 of the revised manuscript).

In the Results the Authors present the total mineral content, however Line 239: “This element is present in fruits in the range of 9.9–94.3 mg/100 g” is it the amount of P in dry matter or fresh product? If it is per fresh product it would be easier to compare values when they are also expressed per dry matter. It is also not clear which fruits? Sweet cherry?

The authors are grateful for the reviewer’s comment. The range values of phosphorus (P) in fruits refers to dry weight of fruits in general and not sweet cherry specifically.

Conclusions

Line 538: “This work allowed a deeper understanding of the mineral content…” Rewrite this sentence. In fact, the Authors analyzed only some of the minerals.

The authors thank the reviewer’s comment. The sentence was reformulated. (Please see now lines 548 and 549 of the revised manuscript).

In my opinion, more precise conclusions are missing. The part (lines 550-552): “may have great potential as food supplements and natural flavoring agents to be used in the pharmaceutical and food industries.” Is very general. More detailed possibilities of using the by-products of sweet cherry flowers, leaves and stems are missing.

The authors thank the reviewer’s comment. As suggested the conclusion section was revised. (Please see now lines 548 to 567 of the revised manuscript)

Round 2

Reviewer 3 Report

Thank you for addressing all the comments and suggestions. In my opinion, the article can be published in a present form.